# Tracheidogram's Classification as a New Potential Proxy in High-Resolution Dendroclimatic Reconstructions

**Mikhail S. Zharkov** [1], **Jian-Guo Huang** [2,3], **Bao Yang** [4], **Elena A. Babushkina** [5], **Liliana V. Belokopytova** [5], **Eugene A. Vaganov** [6,7], **Dina F. Zhirnova** [5], **Victor A. Ilyin** [1], **Margarita I. Popkova** [1] and **Vladimir V. Shishov** [1,3,*]

1   Mathematical Methods and IT Department, Siberian Federal University, Krasnoyarsk 660041, Russia; mzharkov@sfu-kras.ru (M.S.Z.); iljin.victor@gmail.com (V.A.I.); popkova.marg@gmail.com (M.I.P.)
2   Key Laboratory of Vegetation Restoration and Management of Degraded Ecosystems, South China Botanical Garden, Chinese Academy of Sciences, Guangzhou 510070, China; huangjg@scbg.ac.cn
3   Center of Plant Ecology, Core Botanical Garden, Chinese Academy of Sciences, Guangzhou 510070, China
4   Key Laboratory of Desert and Desertification, Northwest Institute of Eco-Environment and Resources, Chinese Academy of Sciences, Lanzhou 730000, China; yangbao@lzb.ac.cn
5   Khakass Technical Institute, Siberian Federal University, Abakan 655000, Russia; babushkina70@mail.ru (E.A.B.); white_lili@mail.ru (L.V.B.); dina-zhirnova@mail.ru (D.F.Z.)
6   Sukachev Institute of Forest, Siberian Branch of the Russian Academy of Science, Krasnoyarsk 660036, Russia; eavaganov@hotmail.com
7   Institute of Ecology and Geography, Siberian Federal University, Krasnoyarsk 660041, Russia
*   Correspondence: vlad.shishov@gmail.com

**Abstract:** Quantitative wood anatomy (QWA) is widely used to resolve a fundamental problem of tree responses to past, ongoing and forecasted climate changes. Potentially, QWA data can be considered as a new proxy source for long-term climate reconstruction with higher temporal resolution than traditional dendroclimatic data. In this paper, we considered a tracheidogram as a set of two interconnected variables describing the dynamics of seasonal variability in the radial cell size and cell wall thickness in conifer trees. We used 1386 cell profiles (tracheidograms) obtained for seven Scots pine (*Pinus sylvestris*) trees growing in the cold semiarid conditions of Southern Siberia over the years 1813–2018. We developed a "deviation tracheidogram" approach for adequately describing the traits of tree-ring formation in different climate conditions over a long-term time span. Based on the NbClust approach and K-means method, the deviation tracheidograms were reliably split into four clusters (classes) with clear bio-ecological interpretations (from the most favorable growth conditions to worse ones) over the years 1813–2018. It has been shown that the obtained classes of tracheidograms can be directly associated with different levels of water deficit, for both the current and previous growing seasons. The tracheidogram cluster reconstruction shows that the entire 19th century was characterized by considerable water deficit, which has not been revealed by the climate-sensitive tree-ring chronology of the study site. Therefore, the proposed research offers new perspectives for better understanding how tree radial growth responds to changing seasonal climate and a new independent proxy for developing long-term detailed climatic reconstructions through the detailed analysis of long-term archives of QWA data for different conifer species and various forest ecosystems in future research.

**Keywords:** radial cell sizes; cell wall thickness; deviation tracheidogram; classification; water deficit; climate reconstructions

## 1. Introduction

There is a fundamental problem of assessing forest reaction to past, ongoing and forecasted climate changes closely connected to the increasing concentrations of greenhouse gases at the middle and high latitudes of the Northern Hemisphere [1–3]. Particularly,

forecasted climate changes could disrupt the functional properties of boreal ecosystems, especially in mountains where plant community restructuring outpaces tree-line advance [2].

Despite the large number of papers concerning tree-ring response to different environmental changes (e.g., temperature increase, significant $CO_2$ trends, additional irrigation, drought effects, etc.), there are no reliable answers concerning how woody plants will respond to climate changes in different forest stands and various biogeographical zones [1,4–10]. As an example, it was shown that falling concentrations of different phytotoxic air pollutants, along with the often cited increasing atmospheric $CO_2$ level, may explain the rising water-use efficiency of trees in forest ecosystems across the Northern Hemisphere over the last two decades [1]. At the same time, most forest species worldwide potentially face long-term significant reductions in productivity if temperature and aridity increase as forecasted across the globe [4].

Quantitative wood anatomy is a good proxy source of encoded information about the environment under climate changes, since it is closely linked to 60% of tree phytomass (stem wood) as a principal pool of carbon sink [11–14]. In 1993, F.H. Schweingruber [15] published a remarkable microphotograph that shows distinct differences in the tree-ring structure of *Pínus ponderosa* growing under optimum and deficit-moisture conditions. The tree-ring anatomy is the final result of several sequential (and possibly overlapping) processes in the seasonal formation of annual rings [16–21]. Where, how and at what stage of seasonal tree-ring formation under changing climate do the differences appear that lead to significant variations in the anatomy of tree rings? Those questions have not yet been definitively resolved. It was shown that there is significant heterogeneity in the spatial patterns of cambium activation for conifer species in the Northern Hemisphere [13], and there are no common climatic triggers to stop a cambium division at the end of the growing season [14]. An intra-annual variability of xylem may not capture the complexity of tree phenological responses to environmental conditions due to the unsuitable but commonly used analytical tools [18]; even in extreme climate conditions, a genetic control may play a role in xylem development [22,23]. Despite the numerous measurements of seasonal growth in woody plants of different species and under different growth conditions (from subarctic to moisture-deficient southern sites), fundamental issues on what determines the simplest anatomical characteristics in conifers (radial cell size and cell wall thickness) and in what intervals of the growing season are still the topic for research discussions [17,24–31].

Several decades ago, it was proposed to estimate the anatomical structure of tree rings in conifers by the tracheidogram approach, which represents the variability of radial cell size and cell wall thickness in the tracheid sequence from the initial cell (the first cell of earlywood) to the final cell (the last cell of latewood) [18,23]. A number of more recent works have shown that the quantitative characteristics of tracheidograms are indicators of the influence of climatic factors on the growth and structure of tree rings at specific growing season intervals [24,26–29,31–36]. Thus, using tracheid traits offers the prospect of greater temporal resolution in dendroclimatic reconstructions [30,36,37]. However, so far, no efforts have been made to consider the tracheidogram as a set of two variables describing the dynamics of seasonal variability in the radial cell size and cell wall thickness, which means that the tracheidogram, as one of integral characteristic of the tree-ring structure, has not yet been used in quantitative anatomy.

In this paper, we have aimed to fill this gap by using the tree-ring anatomical structure as a potential proxy in high-resolution reconstructions. Several objectives have been considered in this work: (a) To develop an adequate method for describing the traits of tree-ring structure to characterize an individual season (year) of growth; (b) to evaluate how similar tracheidograms are in the same calendar years for different trees; (c) to test the hypothesis whether tracheidograms form stable temporal clusters, which in the future can be used to classify the growth seasons whose conditions contribute to the formation of a similar anatomical structure. All the approaches and hypotheses were tested on 200 years of tracheidogram data of Scots pine (*Pinus sylvestris*) growing in semiarid, cold forest-steppe conditions [31].

## 2. Materials and Methods

### 2.1. Study Area

The study was carried out in the eastern part of the Kuznetsk Alatau mountain range in southern Siberia (Figure 1A,B). The altitude ranges from 500 m a.s.l. to 1200 m a.s.l. in the central mountains. A mixed forest comprised of Siberian larch (*Larix sibirica* Ledeb.), Scots pine (*Pinus sylvestris* L.) and silver birch (*Betula pendula* Roth.) in different proportions occupies most of the ridge, but open forest stands alternating with steppe vegetation are found on the drier sites of the southern and southeastern slopes, particularly in the foothills [38].

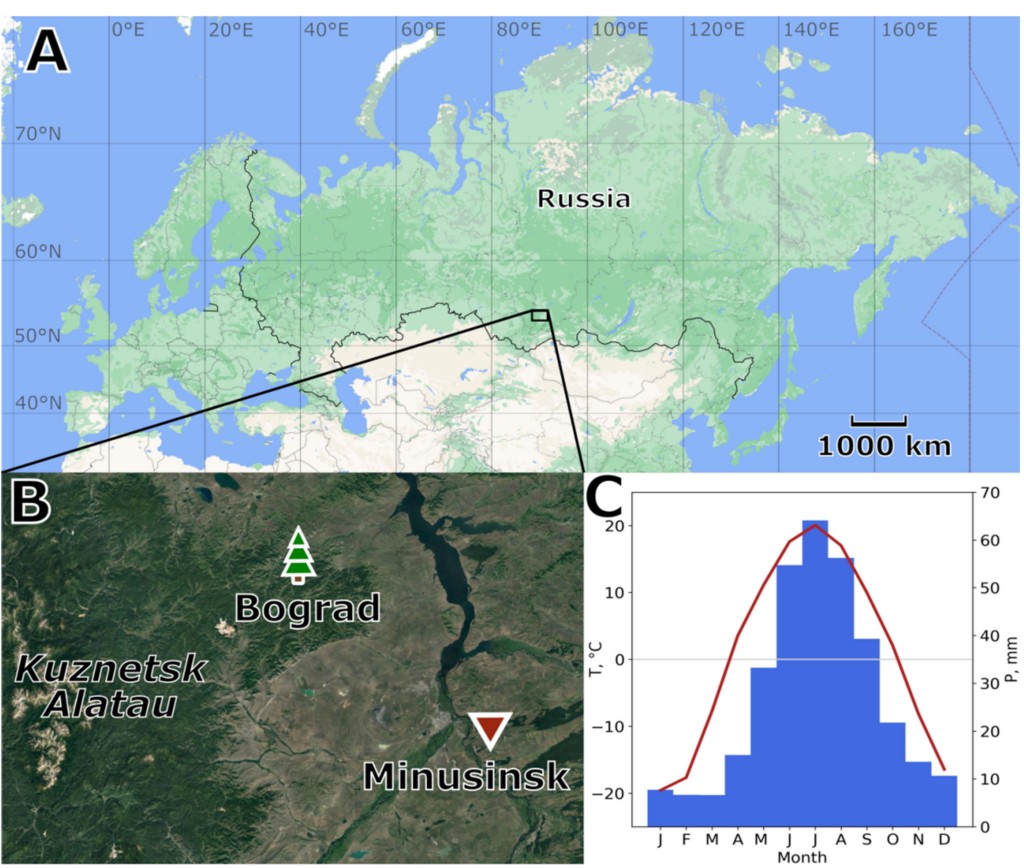

**Figure 1.** The locations of the study plot (green tree) and climate station (red triangle) (**A,B**), and the average monthly mean temperature (°C) and monthly total precipitation (mm) over the years 1917–2018 (**C**).

The region is characterized by the cold, semiarid climate [39,40] having high seasonal and daily temperature variations (Figure 1C). According to the closest meteorological station, Minusinsk (53.68 N, 91.68 E, 251 m a.s.l.), the average annual temperature in the study area is +0.9 °C with an annual precipitation of about 300 mm. Precipitation in the area exhibits an uneven annual distribution, with most of the rainfall (>80%) occurring during the season of positive temperatures, with the highest in July. However, even this amount of precipitation in summer limits the plant growth due to the effect of aridity when the amount of incoming moisture is less than evaporated [40]. The average frost-free interval of positive daily temperatures extends from May to September, where the average daily temperatures are above +5 °C.

### 2.2. Climate Data

The climatic data were collected from the All-Russian Research Institute of Hydrometeorological Information, World Data Center (RIHMIWDC; http://meteo.ru/data/162-

temperature-precipitation (accessed on 18 June 2021)). Particularly, the daily data from the Minusinsk climate station (53.68° N, 91.68° E, 251 m a.s.l.) used covered the years 1917–2018. The temperature was smoothed using a 7-day moving average. The smoothed absolute daily temperature and cumulative precipitation for the tree-ring growing season, estimated there from early May to late September [21,27], were rescaled to relative units in the range [0; 1] by the MinMax transformation [41]. In the work, we considered a cumulative difference between the rescaled temperature and precipitation over the growing season as a square area coefficient (Area) characterizing water deficit.

The monthly self-calibrated Palmer Drought Severity Index (scPDSI) was obtained from the official web UCAR page (http://www.cgd.ucar.edu/cas/catalog/climind/pdsi.html (accessed on 18 June 2021)) and the Climate Explorer web platform (https://climexp.knmi.nl/select.cgi?sc_pdsi (accessed on 18 June 2021)) (for more details, see [42]). The standardized Precipitation Evapotranspiration Index (SPEI) was obtained from the Climate Explorer web platform (https://climexp.knmi.nl/select.cgi?spei_01 (accessed on 18 June 2021)) [43]. Both indexes were used to estimate a sensitivity of long-term tracheid data to the cold, semiarid conditions of the study area.

The bootstrap correlation procedure (number of iterations, 1000) was used to analyze the robustness of relationships between climate and tree-ring traits to avoid false correlations [44].

### 2.3. Sample Collection, Measurement and Tree-Ring Data Processing

Samples of Scots pine wood cores were collected from the vicinity of the Bograd village located at the northeastern tip of the ridge (BGD; 54.20–54.27° N 90.68–90.92° E, 500–650 m a.s.l.) (see [31] for more details). Tree core samples were collected from the upper canopy level of 86 healthy, undamaged adult trees within the forest stand (tree height~10–12 m). Tree-ring width (TRW) series of each sample were measured using the LINTAB-5 system and the TSAP-win program. The quality of cross-dating and measurement accuracy of the individual TRW series were verified using the COFECHA program, which is a component of the dendrochronology program library DPL developed by R. Holmes in 1982 (see details: https://www.ltrr.arizona.edu/software.html (accessed on 18 June 2021)). This classic library allows users to read text files containing tree-ring measurements and use about 30 tools to perform data treatments and statistical analyses included the standardization program ARSTAN, which allows the removal of a non-climate signal from tree-ring series [45]. During the standardization procedure, age-related trends in individual TRW series were removed by 67% cubic smoothing spline with a 50% frequency-response cut-off. Finally, the standard chronology was obtained as a bi-weighted robust average of individual TRW indexes over the years 1812–2018 [45].

### 2.4. Cell Data Processing

Seven cores were selected for anatomical measurements (one core per tree) out of the full sample collection according to maximum cambial age (the number of tree rings from bark to pith), highly significant TRW correlations with the local chronology ($p < 0.01$), and their continuity [31]. A rotary microtome (Microm HM340E; Thermo Fisher Scientific, Indiana, PA, USA) was used to obtain cross-sections of 12–14 μm thickness from the selected tree cores, excluding the areas of juvenile wood (the innermost 10–15 rings of notably different anatomical structure) to minimize height-related trends in wood hydraulic architecture [46]. The cross-sections were stained with safranin and Astra Blue, dehydrated in increasing concentrations of ethanol, washed with xylene, and mounted in Canada balsam on glass slides. Microphotographs of cross-sections were obtained using a digital camera (ProgResGryphax Subra, Jenoptik GmbH, Jena, Germany) mounted on an optical microscope with 200× magnification (BX43, Olympus, Tokyo, Japan). Anatomical traits of all the rings in the selected samples were measured using the semiautomatic program Lineyka 2.01 and ACR-software [47]. Wood anatomical traits, namely, the number of

cells (N), their radial diameter (D), and cell wall thickness (CWT), were measured [48] for five radial cell rows per ring [49].

### 2.5. Statistical Analysis of Tracheidograms

The tracheidograms of the radial cell size and cell wall thickness obtained for the selected trees over the years 1813–2018 were standardized to 15 cells (median value of cell production for the analyzed area over 1813–2018, Figure S1) (see [14] for more details on the standardization procedure).

A seasonal low-frequency trend in the variation of anatomical tree-ring characteristics suppresses high-frequency intra-seasonal variability of the anatomical structure [35]; particularly, the "typical" seasonal dynamics of the radial cell size (i.e., from the largest cells in earlywood to the smallest ones in latewood) suppresses specific intra-seasonal changes in the anatomical characteristics associated with the features of particular tree-ring formation. It can be observed both in the dynamics of the radial cell size (Figure 2A,B) and in the intra-seasonal variability in the cell wall thickness (Figure 2C,D).

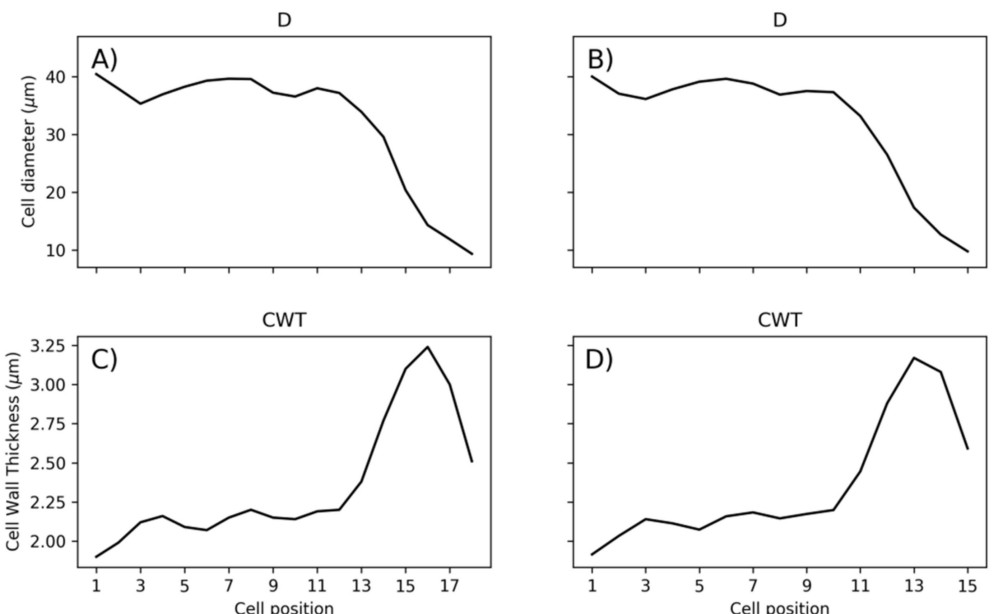

**Figure 2.** Raw (**A**) and standardized to 15 cells (**B**) tracheidogram for the radial cell size D; raw (**C**) and standardized to 15 cells (**D**) tracheidogram for the cell wall thickness CWT for tree 1 in 2000.

To increase the value of intra-seasonal structure features in further statistical assessments of the tracheidograms, we used the "deviation tracheidogram" approach [14,35] (Figure 3).

Two approaches were proposed to obtain deviation tracheidograms (see the Supplementary Materials for details). The first method (Method A) used D and CWT ratios relative to the corresponding general mean tracheidogram for the 7 selected trees (Figure 3), namely: (1) for each year from 1813 to 2018, the average tracheidogram both of D and CWT for the trees was calculated (Figure 3A,C); (2) tree-averaged tracheidograms (206 for D and 206 for CWT) were used to obtain the general mean tracheidogram for D and CWT over 1813–2018 (Figure 3A,C); (3) for each year over 1813–2018, the ratio of the average tracheidogram for D and CWT to the corresponding general mean was considered (Figure 3B,D). The second method (Method B) was based on: (1) the ratio of D and CWT to the corresponding mean annual tracheidogram obtained for each individual tree over 1813–2018; (2) "normalized" tracheidograms for D and CWT of the trees were averaged for each year over 1813–2017.

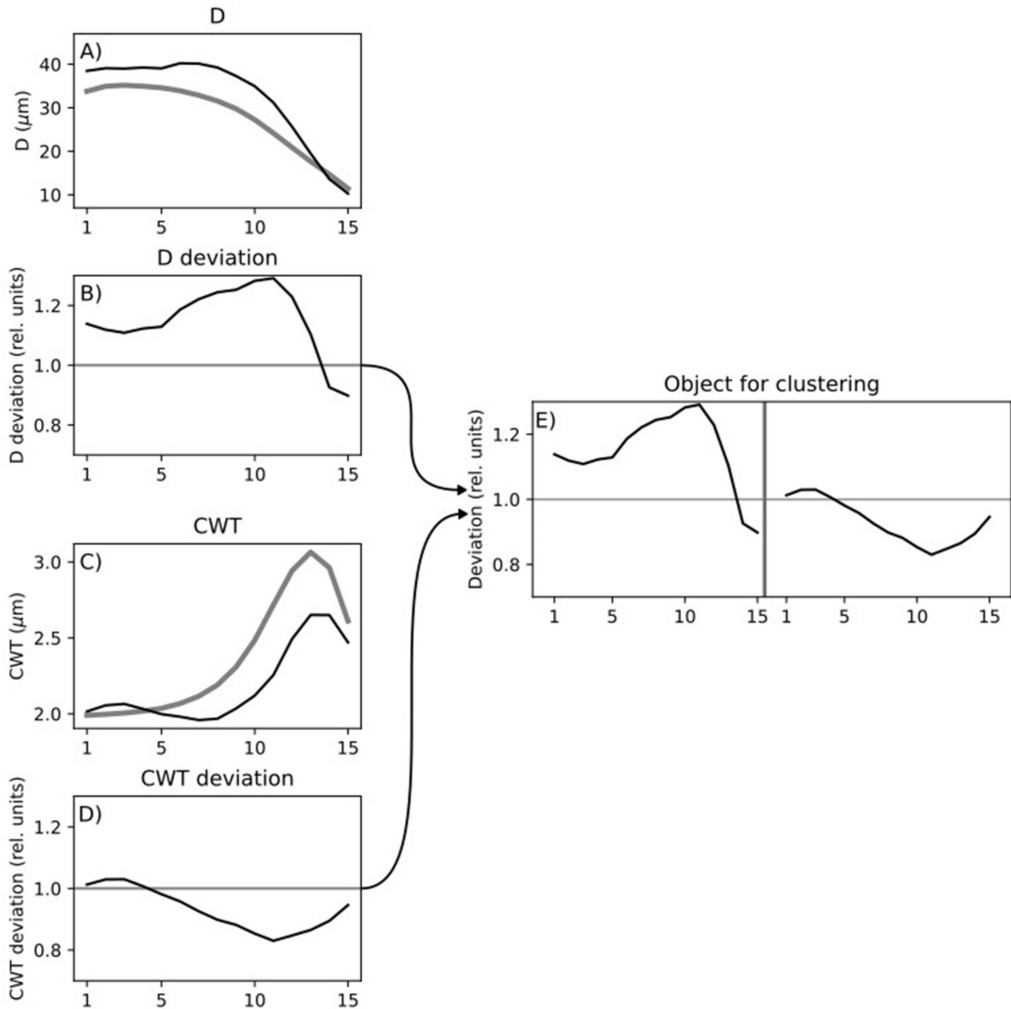

**Figure 3.** Average tracheidogram of the radial cell diameter D for 7 trees in 2000 (black solid curve) and a general mean long-term tracheidogram of D over 1813–2018 (grey solid line) (**A**); relative deviation of the average D tracheidogram in 2000 to the general mean (**B**); average tracheidogram of the cell wall thickness CWT for 7 trees in 2000 (black solid curve) and total mean tracheidogram of CWT over 1813–2018 (grey solid line) (**C**); relative deviation of the mean CWT tracheidogram in 2000 to the total mean (**D**); combined relative deviation for D and CWT (15 values of D and 15 values of CWT considered as a 30-dimensional vector) in 2000 (**E**).

The corresponding code of the algorithm developed using PYTHON (Version 3.9) is available at https://github.com/mikewellmeansme/TracheidClustering (accessed on 18 June 2021).

Deviation tracheidograms, representing 30-dimensional vectors (objects in term of the classification theory) (Figure 3E), were then classified using the NbClust approach to calculate the optimal number of groups (clusters) into which the tracheidograms are discriminated over 1813–2018 (see the description in details https://cran.r-project.org/web/packages/NbClust/NbClust.pdf (accessed on 18 June 2021)), and the K-means method to discriminate tracheidograms (objects) into a predetermined number of classes by the classification error minimization [44].

The Kruskal–Wallis rank test, which is a multivariate generalization of the Mann–Whitney test [50], was applied to check the robustness of the obtained classification. This test allows the detection of statistically significant differences between objects (tracheidograms) of different groups according to their features.

## 3. Results

### 3.1. Classification of Tracheidograms

The deviation tracheidograms obtained by both methods A and B demonstrated highly positive correlations ($p < 0.00001$) for 15 radial D cell sizes and 15 CWT cell wall thicknesses over the years 1813–2018 (Table 1, Figure S1). This is why the deviation tracheidograms obtained by the method A could be used in further analysis.

**Table 1.** Spearman rank correlation coefficients (S) and *p*-values obtained between the deviation tracheidograms obtained by the methods A and B for 15 radial cell size D (D1, D2, D3, ... , D15) and 15 cell wall thickness CWT (CWT1, CWT2, CWT3, ... , CWT15) over 1813–2018.

| D | Spearman | *p*-Value | CWT | Spearman | *p*-Value |
|---|---|---|---|---|---|
| D1 | 0.9972 | $2.0 \times 10^{-227}$ | CWT1 | 0.9906 | $6.9 \times 10^{-175}$ |
| D2 | 0.9983 | $4.1 \times 10^{-248}$ | CWT2 | 0.9918 | $1.7 \times 10^{-180}$ |
| D3 | 0.9987 | $7.0 \times 10^{-262}$ | CWT3 | 0.9917 | $5.3 \times 10^{-180}$ |
| D4 | 0.9990 | $3.6 \times 10^{-273}$ | CWT4 | 0.9901 | $2.1 \times 10^{-172}$ |
| D5 | 0.9991 | $1.4 \times 10^{-275}$ | CWT5 | 0.9883 | $2.5 \times 10^{-165}$ |
| D6 | 0.9990 | $8.0 \times 10^{-272}$ | CWT6 | 0.9893 | $4.1 \times 10^{-169}$ |
| D7 | 0.9986 | $2.3 \times 10^{-257}$ | CWT7 | 0.9878 | $1.5 \times 10^{-163}$ |
| D8 | 0.9985 | $2.4 \times 10^{-253}$ | CWT8 | 0.9894 | $1.3 \times 10^{-169}$ |
| D9 | 0.9968 | $3.8 \times 10^{-221}$ | CWT9 | 0.9889 | $9.6 \times 10^{-168}$ |
| D10 | 0.9981 | $8.9 \times 10^{-245}$ | CWT10 | 0.9876 | $1.2 \times 10^{-162}$ |
| D11 | 0.9970 | $5.8 \times 10^{-224}$ | CWT11 | 0.9914 | $2.0 \times 10^{-178}$ |
| D12 | 0.9966 | $2.2 \times 10^{-218}$ | CWT12 | 0.9923 | $1.9 \times 10^{-183}$ |
| D13 | 0.9986 | $5.3 \times 10^{-258}$ | CWT13 | 0.9944 | $5.1 \times 10^{-197}$ |
| D14 | 0.9978 | $1.7 \times 10^{-237}$ | CWT14 | 0.9959 | $5.8 \times 10^{-211}$ |
| D15 | 0.9920 | $1.6 \times 10^{-181}$ | CWT15 | 0.9970 | $1.4 \times 10^{-224}$ |

It was shown that the optimal discrimination of 206 deviation tracheidograms over 1813–2018 was achieved by separation into four clusters (groups), which was supported by most of the 30 methods used in the comprehensive NbClust approach (Figure 4).

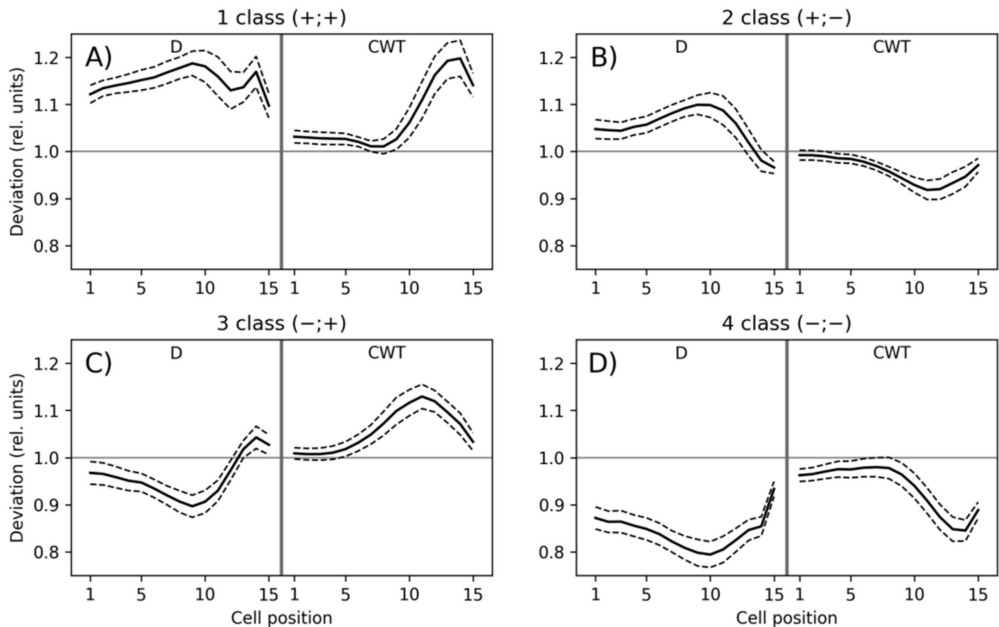

**Figure 4.** 30-dimensional average objects (D and CWT tracheidograms, bold solid curves) for the 1st (**A**), 2nd (**B**), 3rd (**C**) and 4th (**D**) clusters (classes). Plus(+)/Minus(−) indicates the above/below value relative to the cluster (class) average. The dashed curves are ± standard deviation.

We noted that each of the deviation tracheidograms was considered a 30-dimensional vector consisting of 15 D values (features) and 15 CWT features (Figure 3E).

The Kruskal–Wallis test showed that four clusters of tracheidograms are well-identified by all 30 attributes (D1, D2, D3, ... , D15; CWT1, CWT2, CWT3, ... , CWT15) whose contributions to the classification are highly significant ($p < 0.00001$) (Table 2).

**Table 2.** Kruskal–Wallis rank statistics and corresponding *p*-values applied to 30-dimensional tracheidograms (vectors) (D1, D2, D3, ... , D15; CWT1, CWT2, CWT3, ... , CWT15), allocated into four classes.

| D | Statistic | *p*-Value | CWT | Statistic | *p*-Value |
|---|-----------|-----------|-----|-----------|-----------|
| D1 | 116.8515 | $3.67 \times 10^{-25}$ | CWT1 | 46.0809 | $5.45 \times 10^{-10}$ |
| D2 | 127.1252 | $2.25 \times 10^{-27}$ | CWT2 | 40.9941 | $6.55 \times 10^{-9}$ |
| D3 | 133.8806 | $7.88 \times 10^{-29}$ | CWT3 | 34.6134 | $1.47 \times 10^{-7}$ |
| D4 | 143.0068 | $8.49 \times 10^{-31}$ | CWT4 | 34.1818 | $1.81 \times 10^{-7}$ |
| D5 | 145.6427 | $2.29 \times 10^{-31}$ | CWT5 | 39.0002 | $1.73 \times 10^{-8}$ |
| D6 | 152.4864 | $7.66 \times 10^{-33}$ | CWT6 | 43.6884 | $1.75 \times 10^{-9}$ |
| D7 | 158.7064 | $3.48 \times 10^{-34}$ | CWT7 | 49.4108 | $1.06 \times 10^{-10}$ |
| D8 | 158.8744 | $3.20 \times 10^{-34}$ | CWT8 | 57.1523 | $2.38 \times 10^{-12}$ |
| D9 | 155.0766 | $2.11 \times 10^{-33}$ | CWT9 | 74.8697 | $3.86 \times 10^{-16}$ |
| D10 | 147.9417 | $7.32 \times 10^{-32}$ | CWT10 | 93.3626 | $4.15 \times 10^{-20}$ |
| D11 | 129.2131 | $7.99 \times 10^{-28}$ | CWT11 | 114.2045 | $1.36 \times 10^{-24}$ |
| D12 | 113.8614 | $1.61 \times 10^{-24}$ | CWT12 | 135.5797 | $3.39 \times 10^{-29}$ |
| D13 | 113.3792 | $2.05 \times 10^{-24}$ | CWT13 | 141.0537 | $2.24 \times 10^{-30}$ |
| D14 | 121.4426 | $3.77 \times 10^{-26}$ | CWT14 | 138.2426 | $9.04 \times 10^{-30}$ |
| D15 | 85.0263 | $2.56 \times 10^{-18}$ | CWT15 | 128.3065 | $1.25 \times 10^{-27}$ |

Based on the average cluster tracheidograms for D and CWT (Figure 4), the deviation tracheidograms of the first cluster (38 of 202 in total) were characterized by positive deviations from the mean for both radial cell diameter D (+) and cell wall thickness CWT (+) (Figure 4A). The second cluster of tracheidograms (64 of 202) was characterized by positive deviations in D (+) and negative deviations in CWT (−) (Figure 4B). The tracheidograms in the third cluster (48 of 202) were negatively deviated from the mean for D (−) and positively for CWT (+) (Figure 4C). Finally, the tracheidograms of the fourth cluster (58 of 202) were characterized by both negative D (−) and CWT (−) deviations (Figure 4D).

A common tendency can be observed in the variation of the radial cell size from the first to the fourth clusters. As the cluster number increased, the variation of the radial cell size decreased monotonically (Figure 5A). At the same time, the cell wall thickness did not change monotonically. For the first and third clusters, the thickness was significantly higher ($p < 0.05$) than for the second and fourth clusters (Figure 5B).

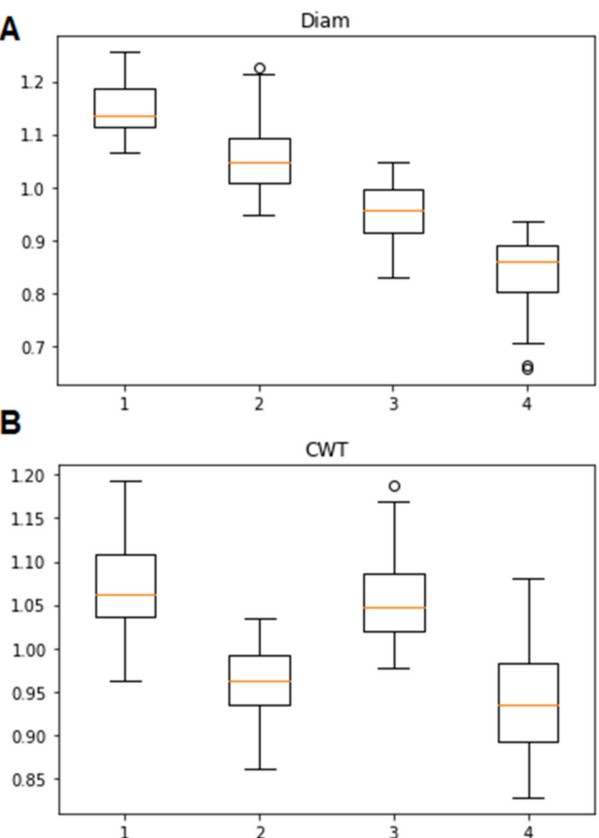

**Figure 5.** Box and Whisker plots for the cell radial diameter Diam (**A**) and cell wall thickness CWT (**B**) for four clusters: orange line—median; box—25% and 75% quartiles; whiskers—non-outlier minimum and maximum; circle—outlier (See the details in [51]).

*3.2. Associations of the Tracheidogram Clusters with Climate Peculiarities*

To compare the tracheidogram clusters with climatic factors, a generalized statistic, namely, the area between the normalized values of seasonal temperature and cumulative precipitation (Area), was used at the first stage (Figure S2). It can be assumed that, as the area increases, the seasonal moisture deficit increases as well.

Considering that each classification object is a tracheidogram linked to a time scale (year), each cluster of tracheidograms was compared with the averaged area obtained for the growing season from May to September over the period of direct climate observations from 1917 to 2018 (Figure S2, Left panel).

The change in the area between the scaled temperature and cumulative precipitation (Area) resulted in a specific response of woody plants to external climatic factors in xylem formation in the current growth season (Figure S2, right panel). Moreover, using the Kruskal–Wallis test, the obtained Areas were significantly different from each other ($p < 0.002$). For the first three clusters of tracheidograms, the median value of the Area also increased with the increasing number (Figure 6).

A bootstrap correlation analysis between standard tree-ring chronology, scPDSI, SPEI, and Area, was calculated for the current growth season from May to September, and tracheidogram clusters (Figure 7) confirmed the significant ($p < 0.01$) influence of moisture limitation (scPDSI, SPEI, and Area) on tree-ring structure (tracheidograms) (Table S1).

There was a highly significant ($p < 0.0001$) negative relationship between the standard tree-ring chronology and tracheidogram classes (Table S1), i.e., an increase in the class number led to a decrease in the growth index value. However, it was shown that there was no significant difference ($p > 0.05$) between the value distributions of tree-ring chronology obtained for two independent time intervals: 1813–1900 and 1901–2017. However, trachei-

dogram class distributions were significantly different ($p < 0.001$) for the same periods. Frequencies of the third and fourth classes over 1813–1900 were significantly higher.

The tracheidogram clusters were significantly negatively correlated with scPDSI and SPEI ($p < 0.001$), but positively correlated with Area ($p < 0.001$) (Table S1). Moreover, scPDSI and SPEI were negatively correlated with Area ($p < 0.001$), which means the greater the area between the scaled temperature and cumulative precipitation, the greater the moisture deficit for the study plot.

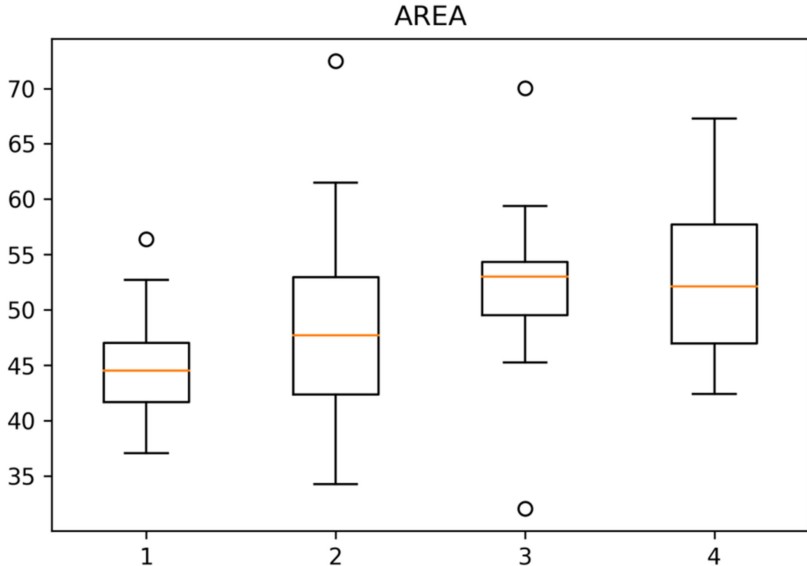

**Figure 6.** Box and Whisker plots for Area corresponding to four classes of tracheidogram: orange line—median; box—25% and 75% quartiles; whiskers—non-outlier minimum and maximum; circle—outlier.

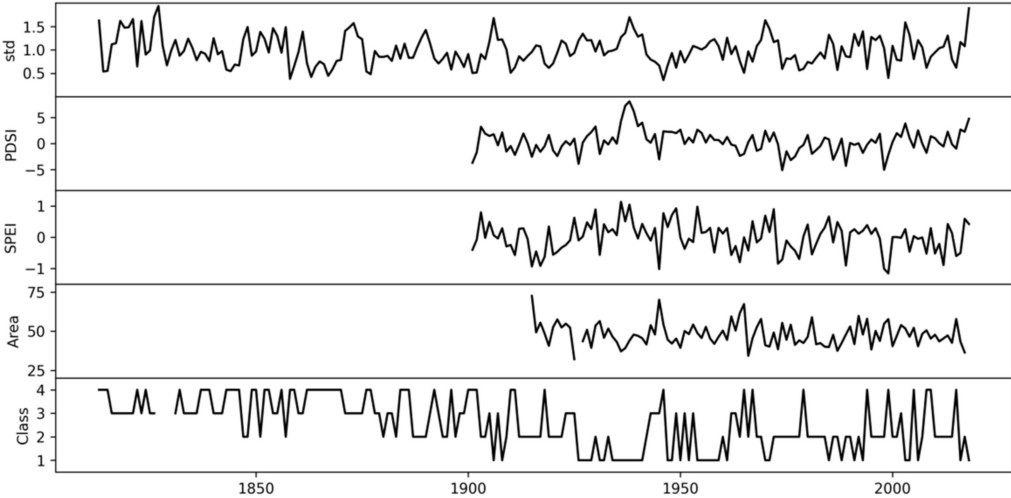

**Figure 7.** Temporal variability of the standard tree-ring chronology, scPDSI, SPEI, Area and tracheidogram classes over 1813–2017.

We noted that the clusters of both scPDSI and SPEI relative to the tracheidograms were also significantly different ($p < 0.005$), but the differences in the median values were not so monotonic as and less significant than in the case of Area (Figure S3).

A comparative analysis of the Kruskal–Wallis test showed highly significant differences (KW = 15.61, $p < 0.0001$) between the third and fourth clusters in the scPDSI values obtained for August and September of the previous year (Figure 8).

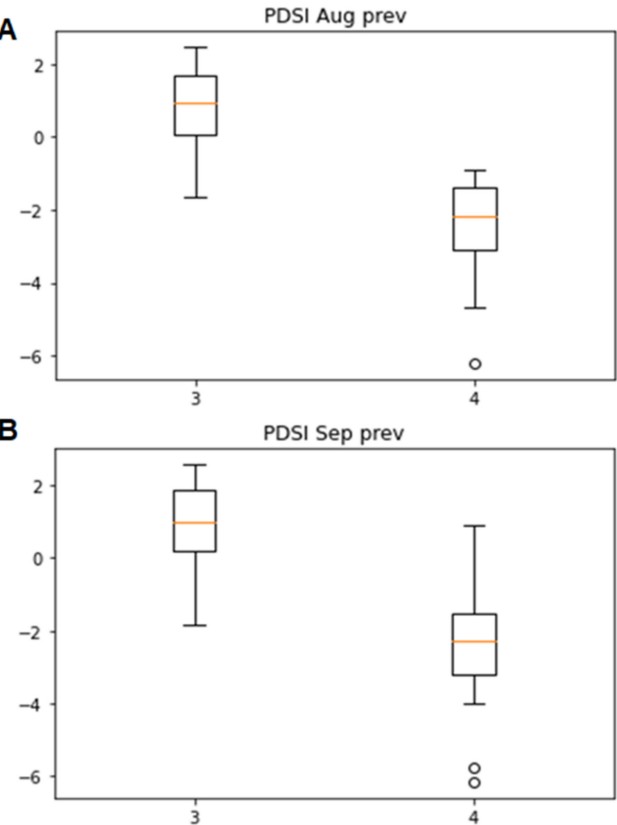

**Figure 8.** Box and Whisker plots for August (**A**) and September (**B**) scPDSI of the previous year corresponding to the third cluster and fourth cluster of tracheidograms: orange line—median; box—25% and 75% quartiles; whiskers—non-outlier minimum and maximum; circle—outlier.

The previous season for the fourth cluster tracheidograms (−;−) was characterized by a significant increase in the moisture deficit compared to the third cluster (−;+), as confirmed by the significantly lower scPDSI values for the fourth cluster (Figure 8).

## 4. Discussion

The use of individual anatomical characteristics for tree rings and their associations with the local climate is presented in previously published papers [36,37,52]. Previously, earlywood and latewood data, the average size of tracheids in earlywood or latewood, cell production, etc., were successfully used [31,53,54]. However, the objectives considered in this work were not formulated or analyzed earlier due to the complexity of the application or the lack of adequate technical and statistical tools [14].

An important component of the work is the development of a methodology (or technique) to use the information encoded in each tracheidogram to analyze the traits of its seasonal formation. It was found that direct measurements of radial sizes and cell wall thickness did not allow the reliable allocation of tracheidograms into clusters, due to the significant amplitude contribution of the seasonal periodic low-frequency component (from larger radial cell sizes at the beginning of the season to smaller ones at the end (Figure 2A,B), and from a smaller cell wall thickness to a larger one (Figure 2C,D)) to the classification results. However, the application of the deviation tracheidogram approach removed the common seasonal tendency of ring formation, and as a result, the classification problem was solved (Figure S1). At the same time, both proposed methods for obtaining deviation tracheidograms (Methods A and B) led to almost identical results (Table 1), which significantly simplified the calculation procedure (https://github.com/mikewellmeansme/TracheidClustering (accessed on 18 June 2021)). There are no doubts that this approach provides new opportunities to maximize the use of the information encoded in tracheidograms.

In the previous work on the analysis of cell chronologies of larch (*Larix gmelini*) trees in the northern boreal forest, three climate clusters corresponding to extremely warm, extremely cold and moderate temperature conditions during the growing season were associated with cell measurements [55]. In this work, we considered a reversed approach. Firstly, we identified clusters of tracheidograms (Figure 4A,B) and, secondly, associated them with the growing season climate conditions characterized by different moisture deficits (Figure S2, Right panel). The first two clusters of tracheidograms were associated with sufficient moisture during most of the growing season (Figures 6 and S2, Right panel), which is reflected in the significant excess of the radial cell size over the mean annual size (Figure 4A,B). Such a trait of tree-ring formation was closely connected with the low values of the coefficient Area, which characterizes moisture deficit (Figure 6), and confirmed by the high values of classical scPDSI and SPEI (Figure S3), which are negatively correlated with Area ($p < 0.01$).

The allocation of tracheidograms into the third and fourth clusters, which were structurally different (Figure 4C,D), cannot be explained by different responses to moisture deficit in the current year (Figure 6). However, the split into the two most "unfavorable" clusters was the result of the significant ($p < 0.01$) moisture deficit in the previous August and September (Figure 8). Such a lagging effect in the moisture deficit of the previous year can be directly connected with a lack of soil moisture at the start of the growing season in the current year, which negatively affects tree-ring formation, especially in the cold, semiarid environment [53]. Moreover, the importance of this effect on tree-ring width and cell production has been reported in many studies for different environments [14,56]. However, the fact that the significant moisture deficit of the end of a previous growing season negatively affects not only the radial cell size, but also the cell wall thickness, requires further detailed research. This effect may appear through the dependence of tree-ring anatomical characteristics (cell production, cell sizes and tree-ring width) on the tree-ring growth rate and, primarily, on cell division in cambium [21,31,57,58].

The proposed classification of tracheidograms allows us to obtain additional time series of anatomical characteristics as a new independent source of climate proxy to the traditionally used reconstruction data (i.e., tree-ring width or maximum density chronologies). In comparison with the tree-ring width chronologies, cell chronologies (i.e., mean cell size, maximum cell wall thickness, position of the transition zone from earlywood to latewood, etc.) [25,30,31] and the obtained cluster chronologies provide a more detailed and comprehensive assessment of past long-term climatic conditions during the growing season (Figure 8). Thus, a higher frequency of growth seasons with significant moisture deficit, corresponding to the third and fourth clusters, were observed over the entire 19th century compared to the 20th century (Figure 7). Moreover, the 20th century is characterized by a full range of moisture regimes that are responsible for the formation of the corresponding tracheidograms (Figure 7), whereas the variability of the traditional tree-ring chronology, most commonly used for climatic reconstructions [23], had no peculiarities over the entire time span. This diversity of the classified seasons is consistent with the increasing climate "instability" noted by many studies in recent decades [59,60]. Moreover, the climate in west China and central Asia (including southern Siberia) is changing from warm and dry to extremely uneven fluctuations in water availability during the past three decades (1987–2014), especially in the mountain regions (Tibetan Plateau, West Qinling Mountains, Altai and Sayan Mountains) [59,60], so it is critical to develop new approaches (as the proposed tracheidogram classification) to obtain a more accurate understanding of the climate—tree growth relationships, process of ring formation and more detailed climate reconstructions.

## 5. Conclusions

This work proposes a new approach to classifying tree rings based on their structure, where anatomical traits are considered as an interconnected set of two characteristics: radial cell size and cell wall thickness over more than 200 years (1813–2018). In the framework

of the approach, the deviation tracheidogram procedure was developed by which the common seasonal tendency of ring formation (from larger radial cell sizes at the beginning of the season to smaller ones at the end) is removed, and as a result, the transformed tracheidograms reliably allocate into clusters.

The work demonstrated that the deviation tracheidograms obtained for seven trees reliably split into four clusters with a clear biological and ecological interpretation over a long-term time span, 1813–2018 (Figure 4). Thus, the median radial cell size decreased significantly with the increasing cluster number (Figure 5A), while the thickness of the cell wall did not follow a monotonic pattern (Figure 5B). Whether this result is sufficiently general or specific to the species analyzed or to the local conditions of the plot is a challenging issue for further research.

The obtained classification of tracheidograms can be associated with different climate conditions, particularly with different levels of water deficit for both the current and previous growing seasons. However, the fact that the significant moisture deficit of the previous growing season negatively affects both the radial cell size and the cell wall thickness requires further detailed research for other species and habitats.

The proposed classification of tracheidograms allows us to obtain a new, independent source of climate proxy with higher temporal resolution than the traditionally used reconstruction data. The obtained cluster reconstruction shows that the entire 19th century was characterized by significant water deficit, which was not revealed by the climate-sensitive tree-ring width chronology of the study site (Figure 7).

Obviously, the proposed approach and obtained results can be used for better understanding how tree radial growth responds to changing seasonal climate through the detailed analysis of long-term archives of wood anatomy data for different conifer species and various forest ecosystems in future research.

Therefore, the proposed approach of tracheidogram classification offers new perspectives in developing long-term detailed climatic reconstructions and in spatiotemporal comparisons of the growing season climate conditions and their association with the observed and forecasted regional climatic trends.

**Supplementary Materials:** The following supporting information can be downloaded at: https://www.mdpi.com/article/10.3390/f13070970/s1, Mathematical description of the deviation tracheidogram approach; Table S1: Bootstrapped correlations between standard tree-ring chronology (std), residual chronology (res), scPDSI (Mean_PDSI) and SPEI (Mean_SPEI), averaged from May to September, square area obtained as a difference between scaled temperature and precipitation (Area), classes of tracheidograms (Class); Figure S1: Deviation tracheidograms obtained by Method A (left panel) and Method B (central panel) and their differences (right panel) over 1813–2018; Figure S2: Temperature (red curve) and cumulative precipitation (blue curve) 7-day moving average over the growing season (from 1 May to 30 September) in absolute values (Left Panel) and the climate factors scaled by the MinMax approaches in relative units (Right Panel) for the years with the median Area in the corresponding Class. The dashed grey area is the area of differences between the rescaled temperature and precipitation over the growing season. The number above the dashed area is the value of Area statistics for the corresponding year; Figure S3: Box and Whisker plots for scPDSI (A) and SPEI (B) corresponding to four classes of tracheidogram: orange line—median; box—25% and 75% quartiles; whiskers—non-outlier minimum and maximum.

**Author Contributions:** V.V.S., E.A.V. and M.S.Z. designed this study. E.A.B., L.V.B. and D.F.Z. performed fieldwork, collected and obtained wood anatomy data. M.S.Z., V.A.I. and V.V.S. performed data analysis. J.-G.H. and B.Y. contributed with valuable comments, new ideas and discussion. M.I.P. performed English proofreading. All authors wrote the manuscript. All authors have read and agreed to the published version of the manuscript.

**Funding:** This research was funded by the Russian Ministry of Science and Higher Education (# FSRZ 2020-0010—data collection) and the Russian Science Foundation (RSF#18-14-00072 P—data preparation and treatment; RSF#22-14-00048—developing the algorithms and program codes).

**Institutional Review Board Statement:** Not applicable.

**Informed Consent Statement:** Not applicable.

**Data Availability Statement:** Not applicable.

**Acknowledgments:** We really appreciated both reviewers for their constructive criticism and recommendations to improve the MS.

**Conflicts of Interest:** The authors declare no conflict of interest.

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
