# Peer review of "Tracheidogram’s Classification as a New Potential Proxy in High-Resolution Dendroclimatic Reconstructions"

_forests, doi:10.3390/f13070970_

Round 1

Reviewer 1 Report

This research encompasses a lot of work.  It’s tedious enough measuring ring widths individually across many 10s of 200-year old specimens, but measuring radial length of tracheids within each ring and cell wall thicknesses multiplies the amount of work of dendrochronology by at least 100 times.  I hope it’s worth it.

Some comments on this ms, in order of reading through it:

Introduction: The Introduction starts with three paragraphs that are each one sentence long.  It seems as if each sentence should be the topic sentence of a longer paragraph (additional sentences that illustrate to topic).

Line 53: Pinus ponderósa.  The tilde over the o isn’t necessary, or even an official spelling of that species name.

Line 88: Kuznetsk Alatau mountain range (Figure 1)?  This geographical name isn’t included in the map of Figure 1.

Figure 1: Speaking of these maps, a regional perspective is needed.  And, can the mean annual total precipitation be give in C?  Anticipating that summer moisture availability might be a limiting factor to tree growth in this research, it seems from this climograph that it rains a lot at this site during the growth season.

Line 115: (Figure 6) already?  Before Figures 2-5?  Seems odd.

Lines 133-134: Not much detail is given about COFECHA or ARSTAN.  Presumably dendro specialists know these programs well, but will all readers of Forests know them?

Table 1: Is the distinction between Methods A and B really important?  By these data in Table 1, they appear to yield identical results, as stated later in the text (line 321): “Methods A and B lead to almost identical results (Table 1).”

Figure 6: At first glance, these plots look quite similar to one another across classes.  And yet, the text says the obtained areas are significantly different from one another (line 266).  Trusting that statistical significance was obtained, the question emerges: are differences between classes truly important?

Figure 8: This figure has elements of traditional dendroclimatology, i.e., annual time-series of the standard tree-ring chronology and selected climate variables.  Normally, time-series like these are overlay plotted, making it easier to see their correlation or lack thereof.  Either way, it is also common to include horizontal reference lines on long time-series plots, making it easier to pick out years or periods that are unusual.

Lines 275-276: “Figure 8 confirmed the significant influence of moisture limitation on tree rings.”  OK, but again, from Figure 1C, showing ample rainfall being typical during the growing season, this finding seems counter intuitive.

Lines 280-282: “There is a highly significant negative relationship between standard tree-ring chronology and tracheidogram classes, i.e., an increase in class number leads to a decrease in growth index value.”  Presumably a significant relationship is a favorable result, but this begs the question: Does this tracheidogram approach truly add new information above and beyond traditional dendrochronology techniques?  For all the extra work it takes to develop tracheidograms, they should yield information above and beyond traditional dendro using plain old ring widths.

60 refs cited.  Good.

Author Response

This research encompasses a lot of work.  It’s tedious enough measuring ring widths individually across many 10s of 200-year old specimens, but measuring radial length of tracheids within each ring and cell wall thicknesses multiplies the amount of work of dendrochronology by at least 100 times.  I hope it’s worth it.

Dear Reviewer, thank you very much for the positive feedback and constructive comments to improve the manuscript. Indeed, we done a lot of work in cell measurements, which are now being simplified due to various semi-automatic tools developed (ROXAS, ACR, Lineyka, etc). We hope that such measurements can be useful in obtaining new climate reconstructions due to the higher temporal resolution (from year-to year to intra-annual scale) and new analytical algorithms.

Some comments on this ms, in order of reading through it:

Introduction: The Introduction starts with three paragraphs that are each one sentence long.  It seems as if each sentence should be the topic sentence of a longer paragraph (additional sentences that illustrate to topic).

We added additional sentences and reformatted the paragraphs, respectively

Line 53: Pinus ponderósa.  The tilde over the o isn’t necessary, or even an official spelling of that species name.

We corrected it respectively

Line 88: Kuznetsk Alatau mountain range (Figure 1)?  This geographical name isn’t included in the map of Figure 1.

We rebuilt the map and added ‘Kuznetsk Alatau’

Figure 1: Speaking of these maps, a regional perspective is needed.  And, can the mean annual total precipitation be give in C? 

We corrected the figure capture:1. The locations of the study plot (green tree) and climate station (red triangle) (A, B) and average monthly mean temperature (oC) and monthly total precipitation (mm) over 1917-2018 (C).”

Anticipating that summer moisture availability might be a limiting factor to tree growth in this research, it seems from this climograph that it rains a lot at this site during the growth season.

Thank you very much for the question. We clarified this issue in the text:

“Precipitation of the area exhibited an uneven annual distribution with most of the rainfall (>80%) occurring during the season of positive temperatures, with the highest in July. But even this amount of precipitation in summer time limits the plant growth due to the effect of aridity when the amount of incoming moisture is less than evaporated [40].”

Line 115: (Figure 6) already?  Before Figures 2-5?  Seems odd.

We deleted the link on Figure 6.

Lines 133-134: Not much detail is given about COFECHA or ARSTAN.  Presumably dendro specialists know these programs well, but will all readers of Forests know them?

We added an additional information in the paragraph:

“The quality of cross-dating and measurement accuracy of the individual TRW series were verified using the COFECHA program which is a component of dendrochronology program library DPL developed by R. Holmes in 1982 (see in details: https://www.ltrr.arizona.edu/software.html). This classic library allows users to read text files containing tree-ring measurements and perform about 30 tools of data treatments and statistical analyses included the standardization program ARSTAN which allows to remove a non-climate signal from tree-ring series [45]. During the standardization procedure, age-related trends in individual TRW series were removed by 67% cubic smoothing spline with a 50% frequency-response cut-off . Finally, the standard chronology was obtained as a bi-weighted robust average of individual TRW indexes over 1812-2018 [45].”

Table 1: Is the distinction between Methods A and B really important?  By these data in Table 1, they appear to yield identical results, as stated later in the text (line 321): “Methods A and B lead to almost identical results (Table 1).”

Two methods are not mathematically identical (see formal mathematical description https://github.com/mikewellmeansme/TracheidClustering/blob/master/method_description.md). We added the mathematical description in the SM. We are doubt that similarity in the results (Table 1) can be obtained for other species and habitats. Additional testing is needed. This is why we highlighted  distinction between Methods A and B

Figure 6: At first glance, these plots look quite similar to one another across classes.  And yet, the text says the obtained areas are significantly different from one another (line 266).  Trusting that statistical significance was obtained, the question emerges: are differences between classes truly important?

The difference between ‘climatic’ classes are very important in the context of this work. Firstly, we identified clusters of tracheidograms which are significantly different by wood anatomy structure (Figure 4) and by statistics (Figure 5) over the last two centuries. Then we tried to associate them with the growing season climate conditions characterized by different moisture deficits based on direct climate observations. We considered four ‘climatic’ classes (Figure 6, Right panel) and shown these classes are statistically different (Figure 7). By this way we shown that the different moisture deficit plays a significant role in the formation of specific tree-ring structure reflecting in the clusters of tracheidograms 

We are agreed the differences between climatic classes are not visually obvious on the figure 6 but they are highly significant on the figure 7. We corrected the figure by introducing the Area value of each ‘climatic’ classes. We relocated the figure 6 in SM section.Figure 8: This figure has elements of traditional dendroclimatology, i.e., annual time-series of the standard tree-ring chronology and selected climate variables.  Normally, time-series like these are overlay plotted, making it easier to see their correlation or lack thereof.  Either way, it is also common to include horizontal reference lines on long time-series plots, making it easier to pick out years or periods that are unusual.

Lines 275-276: “Figure 8 confirmed the significant influence of moisture limitation on tree rings.”  OK, but again, from Figure 1C, showing ample rainfall being typical during the growing season, this finding seems counter intuitive.

The study region is characterized as a cold semi-arid territory, where annual precipitation amount (about 300 mm) is comparable with dry regions in Northern Hemisphere, e.g. some parts of Northern Mongolia, Tibetan Plato, etc., where tree growth is strongly limited by water deficit. It was shown early (Shah et al., 2015) that even 80% of rainfall in growing season (particualry, June-July-August precipitation) is not enough for comfortable tree-ring growth due to the ‘effect of aridity’.

We clarified this issue in the text: (See Lines 119-121)

“Precipitation of the area exhibited an uneven annual distribution with most of the rainfall (>80%) occurring during the season of positive temperatures, with the highest in July. But even this amount of precipitation in summer time limits the plant growth due to the effect of aridity when the amount of incoming moisture is less than evaporated [40].”

Lines 280-282: “There is a highly significant negative relationship between standard tree-ring chronology and tracheidogram classes, i.e., an increase in class number leads to a decrease in growth index value.”  Presumably a significant relationship is a favorable result,

We are agreed those correlation is a result of mathematical manipulations, i.e. inversed numbering of the classes will result to positive relationship. But what is important was added to the Lines 301-305:

However, it was shown there is no significant difference (p>0.05) between value distributions of tree-ring chronology obtained for two independent time intervals: 1813-1900 and 1901-2017. But tracheidogram class distributions are significantly different (p<0.001) for the same periods. Frequencies of 3rd and 4th classes are significantly higher over 1813-1900.

but this begs the question:

Does this tracheidogram approach truly add new information above and beyond traditional dendrochronology techniques?  For all the extra work it takes to develop tracheidograms, they should yield information above and beyond traditional dendro using plain old ring widths.

What is really important in our approach are the following points of our research which we highlighted in Discussion (Lines 373-395)

“The proposed classification of tracheidograms allows us to obtain additional time series of anatomical characteristics as a new independent source of climate proxy to the traditional reconstruction-used data (i.e., tree-ring width or maximum density chronologies). In comparison with the tree-ring width chronologies, cell chronologies (i.e., mean cell size, maximum cell wall thickness, position of the transition zone from earlywood to latewood, etc.) [25,30,31] the obtained cluster chronologies provide a more detailed and comprehensive assessment of past long-term climatic conditions during the growing season (Figure 8). Thus, a higher frequency of growth seasons with significant moisture deficit, corresponding to the 3rd and 4th clusters were observed over the whole 19th century compared to the 20th century (Figure 8). Moreover, the 20th century is characterized by a full range of moisture regimes which are responsible for the formation of the corresponding tracheidograms (Figure 8). Whereas, the variability of the traditional tree-ring chronology, most commonly used for climatic reconstructions [23], had no peculiarities over the whole time span. This diversity of the classified seasons is consistent with the increasing climate "instability" noted by many studies in recent decades [59,60]. Moreover,  the climate in west China and central Asia (including southern Siberia) is changing from warm dry to extremely uneven fluctuations in water availability  during the past 3 decades (1987-2014), especially in the mountain regions (Tibetan Plateau, West Qinling Mountains, Altai and Sayan Montains) [59,60], so it is critical to develop new approaches (as the proposed tracheidogram classification) to get more accurate understanding of the climate - tree growth relationships, process of ring formation and more detailed climate reconstructions.”

and Conclusion (Lines 417-421):

“The proposed classification of tracheidograms allows us to obtain a new independent source of climate proxy with higher temporal resolution to the traditional reconstruction-used data. The obtained cluster reconstruction shows that entire 19th century was characterized by significant water deficit which was not revealed by the climate sensitive tree-ring width chronology of the study site (Figure 8).”

60 refs cited. Good.

Thank you very much

Reviewer 2 Report

The paper is interesting and probably brings very novel know-how in the field of dendrochronology and dendroclimatology. On the other hand, the text and also partly structure of the manuscript would be improved. I recommend major revision of the manuscript.  

 Specific comments and recommendations:  

 Abstract

I feel like the Abstract is the main weakness of the paper. It is too general, missing some more information about the obtained results. Moreover, some sentences are just repeating statements in the same way as were shown in the other sections. For instance, the first sentence of the Abstract section is the exactly the same as the one on in the very beginning of the Introduction section. Then, the last sentence of the Abstract section is completely the same as the one in the end of the Discussion section etc. The text of the Abstract section must be deeply remade!

 Introduction

Line 49-51. The idea of this sentence does not sound me logical. The authors would reconsider what they are attempting to state.

Line 58. The sentence related to the cites 13, 14, 18, 22, and 23 is very vague or unclear to me. It would be fine to explain more carefully in which context the five cited papers showed up that “the questions have not yet been definitively resolved”!

 Material and Method  

Figure 1. I do not like the subfigure A. The readers outside Russia do not know much about the region of Siberia. Thus, the subfigure A would show broader scale to understand position of the experimental plot. Moreover, the authors might indicate location of the Minusinsk climatic station (in the subfigure B).

Line 115 – do not refer the Figure 6. It is not in the order with other figures and in addition, the information in the figure would be a part of the Results section.   

Figure 2., Figure 3., Table .1, Figure 4., - all these figures and table as well as the related text would be a part of the Results section!   

Table 1 and Table 2 – use decimal dots, please.

Results

The section is nearly OK.

Figure 6 – give the name also for the y-axis in the left column of the figures (e.g. Climate factors in relative units).

Discussion

The section is rather fine. However, I believe that the authors would add also a Conclusion section. Then, some part of the current Discussion text would be moved to the new Conclusion section. I mean especially strong generalization of the new method, comparisons with the previous procedures and techniques, its advantages and weaknesses, further prospect for real implementation in dendroclimatology. Possibly also unclear issues which would be solved in the future research might be explained here

Author Response

The paper is interesting and probably brings very novel know-how in the field of dendrochronology and dendroclimatology. On the other hand, the text and also partly structure of the manuscript would be improved. I recommend major revision of the manuscript.

Dear Reviewer, thank you very much for the positive feedback and constructive comments to improve the manuscript. We hope that our approach will be useful for the experts in digital wood anatomy along with dendroclimatologists and dendrochronologists taking into account massive application of those data in eco-climatic research worldwide. We corrected our MS followed by the all your recommendations

Specific comments and recommendations: 

 Abstract

I feel like the Abstract is the main weakness of the paper. It is too general, missing some more information about the obtained results. Moreover, some sentences are just repeating statements in the same way as were shown in the other sections. For instance, the first sentence of the Abstract section is the exactly the same as the one on in the very beginning of the Introduction section. Then, the last sentence of the Abstract section is completely the same as the one in the end of the Discussion section etc. The text of the Abstract section must be deeply remade!

We revised the abstract, respectively: (Lines 19-38)

Abstract: Quantitative wood anatomy (QWA) is widely used to resolve a fundamental problem of tree responses to the past, ongoing and forecasted climate changes. Potentially QWA data can be considered as a new proxy source in long-term climate reconstruction with higher temporal reso-lution to traditional dendroclimatic data. In this paper we considered a tracheidogram as a set of two interconnected variables describing the dynamics of seasonal variability in the radial cell size and cell wall thickness for the conifer trees. We used 1386 cell profiles (tracheidograms) obtained for seven Scots pine (Pinus sylvestris) trees growing in the cold semiarid conditions of Southern Siberia over 1813-2018. We developed a ‘deviation tracheidogram’ approach for adequately describing the traits of tree-ring formation in different climate conditions over a long-term time span. Based on the NbClust approach and K-means method the deviation tracheidograms were reliably split into four clusters (classes) with clear bio-ecological interpretations (from the most favorable growth conditions to worse ones) over 1813-2018. It was shown that the obtained classes of tracheidograms can be directly associated with different levels of water deficit both for the current and previous growing seasons. The tracheidogram cluster reconstruction shows that the entire 19th century was characterized by considerable water deficit which was not been revealed by the climate sensitive tree-ring chronology of the study site. Therefore, the proposed research offers new perspectives for better understanding how tree radial growth responds to changing seasonal climate and a new independent proxy in developing long-term detailed climatic reconstructions through the detailed analysis of long-term archives of QWA data for different conifer species and various forest eco-systems in future research.

Introduction

Line 49-51. The idea of this sentence does not sound me logical. The authors would reconsider what they are attempting to state.

We clarified our statement in the text: (Lines 53-58)

“Despite the large number of papers concerning tree-ring response to different environmental changes (i.e., temperature increase, significant CO2 trends, additional irrigation, drought effects, etc.) there are no reliable answers how woody plants will respond to climate changes in different forest stands and various biogeographical zones [1,4–10]. As an example, it was shown that falling concentrations of different phytotoxic air pollutants along with most cited increasing atmospheric CO2 level may explain rising water-use efficiency  of trees in forest ecosystems across Northern Hemisphere over the last two decades [1]. At the same time most of forest species worldwide potentially face long-term significant reductions in productivity if temperature and aridity increase as forecasted across the globe [4].”

Line 58. The sentence related to the cites 13, 14, 18, 22, and 23 is very vague or unclear to me. It would be fine to explain more carefully in which context the five cited papers showed up that “the questions have not yet been definitively resolved”!

We clarified the issue in the MS: (Lines 67-74)

“Those questions have not yet been definitively resolved. So it was shown that there are significant heterogeneity in the spatial patterns of cambium activation for conifer species in Northern Hemisphere [13], there are no common climatic trigger to stop a cambium division at the end of growing season [14] , an intra-annual variability of xylem may not capture the complexity of tree phenological responses to environmental conditions due to the unsuitable but commonly used analytical tools [18], even in extreme climate conditions a genetic control may play a role in xylem development [22, 23] .”

 Material and Method 

Figure 1. I do not like the subfigure A. The readers outside Russia do not know much about the region of Siberia. Thus, the subfigure A would show broader scale to understand position of the experimental plot. Moreover, the authors might indicate location of the Minusinsk climatic station (in the subfigure B).

We rebuilt the map, respectively.

Line 115 – do not refer the Figure 6. It is not in the order with other figures and in addition, the information in the figure would be a part of the Results section.

Done.

Figure 2., Figure 3., Table .1, Figure 4., - all these figures and table as well as the related text would be a part of the Results section!

The figures 2 and 3 are visual explanations for better understanding of wider audience how the described algorithms process the wood anatomy data used. They can be considered as a part of methods used.

Table 1 and figure 4 are moved to the Results.

Table 1 and Table 2 – use decimal dots, please.

Corrected.

Results

The section is nearly OK.

Figure 6 – give the name also for the y-axis in the left column of the figures (e.g. Climate factors in relative units).

We corrected the figure, respectively and moved it in the Supplementary Materials

Discussion

The section is rather fine. However, I believe that the authors would add also a Conclusion section. Then, some part of the current Discussion text would be moved to the new Conclusion section. I mean especially strong generalization of the new method, comparisons with the previous procedures and techniques, its advantages and weaknesses, further prospect for real implementation in dendroclimatology. Possibly also unclear issues which would be solved in the future research might be explained here

Thank you very much for the recommendation to restructure the final part of the MS! We introduced a Conclusion following all your suggestions: (Lines 397-429)

“5. Conclusion

This work proposes a new approach to classifying tree rings based on their structure, where anatomical traits are considered as an interconnected set of two characteristics: radial cell size and cell wall thickness over more than 200 years (1813-2018). In the framework of the approach the deviation tracheidogram procedure was developed by which the common seasonal tendency of ring formation (from larger radial cell sizes at the beginning of the season to smaller ones at the end) is removed and as a result, the transformed tracheidograms reliably allocate into clusters.

The work demonstrated that the deviation tracheidograms obtained for 7 trees are reliably split into 4 clusters with a clear biological and ecological interpretation over a long-term time span 1813-2018 (Figure 4). Thus, the median radial cell size decreases significantly with the increasing cluster number (Figure 5A), while the thickness of the cell wall does not follow a monotonic pattern (Figure 5B). Whether this result is sufficiently general or specific to the species analyzed or to the local conditions of the plot is a challenging issue for further research.

The obtained classification of tracheidograms can be associated with different climate conditions, particularly with different levels of water deficit both for the current and previous growing seasons. But the fact that the significant moisture deficit of the previous growing season negatively affects both the radial cell size and the cell wall thickness requires further detailed research for other species and habitats.

The proposed classification of tracheidograms allows us to obtain a new independent source of climate proxy with higher temporal resolution to the traditional reconstruction-used data. The obtained cluster reconstruction shows that entire 19th century was characterized by significant water deficit which was not revealed by the climate sensitive tree-ring width chronology of the study site (Figure 8).

Obviously, the proposed approach and obtained results can be used for better understanding how tree radial growth responds to changing seasonal climate through the detailed analysis of long-term archives of wood anatomy data for different conifer spe-cies and various forest ecosystems in future research.

Therefore, the proposed approach of tracheidogram classification offers new perspectives in developing long-term detailed climatic reconstructions, in spatio-temporal comparison of the growing season climate conditions and their association with the observed and forecasted regional climatic trends.”

Round 2

Reviewer 2 Report

I could see the reactions of the authors to my comments I feel like all of them were respected and fully incorporated in the msc.